# Fused Orthogonal Alternating Least Squares for Tensor Clustering

**Jiacheng Wang**
Department of Statistics
University of Chicago
jiachengwang@uchicago.edu

**Dan Nicolae**
Department of Statistics
University of Chicago
nicolae@statistics.uchicago.edu

## Abstract

We introduce a multi-mode tensor clustering method that implements a fused version of the alternating least squares algorithm (Fused-Orth-ALS) for simultaneous tensor factorization and clustering. The statistical convergence rates of recovery and clustering are established when the data are a noise contaminated tensor with a latent low rank CP decomposition structure. Specifically, we show that a modified alternating least squares algorithm can provably recover the true latent low rank factorization structure when the data form an asymmetric tensor with perturbation. Clustering consistency is also established. Finally, we illustrate the accuracy and computational efficient implementation of the Fused-Orth-ALS algorithm by using both simulations and real datasets.

## 1 Introduction

Tensors, or multidimensional arrays, have played an essential role in a wide range of scientific and business applications, including neuroscience [9, 16], social networks [18, 4] and recommendation systems [3, 28]. As higher order tensors have been extensively used as a framework for storing and organizing massive data, the associated need to develop methods for tensor multi-mode clustering has sharply increased. Several methods have been proposed for matrix biclustering for columns and rows simultaneously [6, 5, 7, 22]. However, those methods cannot be directly applied to tensors due to the complex higher order generalizations of the matrix singular value decomposition (SVD) and principal component analysis (PCA). Motivated by the methodology gap between low and high dimensional arrays, we investigate here the multi-mode clustering problem for tensors.

We propose the Fused Orthogonal Alternating Least Squares (Fused-Orth-ALS) algorithm for uncovering co-clustering structures in high order tensors. Clustering along a specific mode is equivalent to clustering the slice matrices for that mode as shown in Figure 1. The core idea is to model true cluster means as having a rank-1 matrix basis. The method is designed to systematically investigate and achieve the following. First, under mild assumptions on the tensor decomposition structure, our approach encourages smoothness in the decomposed components, leading to clustering performance with higher accuracy. Second, orthogonality is pursued for the decomposed components, avoiding the problem of local-minima convergence of the classical ALS tensor decomposition algorithm, preventing multiple recoveries of the same factors, and achieving faster convergence rates compared to popular tensor decomposition methods. Third, theoretical guarantees are provided for the recovery and clustering consistency with even a single tensor sample, which is difficult to achieve in vector or matrix clustering. The results in synthetic and real data demonstrate the robustness of our approach under mild model misspecification and assumption violations, and the effectiveness under high dimensional settings for extracting underlying clustering structures. Even though we focus on order three tensors, it is straightforward to derive analogous results for higher order tensors.

36th Conference on Neural Information Processing Systems (NeurIPS 2022).

**Related work.** Multiple papers are closely related to, but also clearly distinct from, our proposed methodology for tensor multi-mode clustering. Tensor clustering utilizing tensor block model appears in Chi et al. [8] and is further developed in Wang and Zeng [26]. A popular alternative to the tensor block model is the CANDECOMP/PARAFAC (CP) decomposition that captures structure using the sum of rank-1 tensors. Our paper adopts the CP decomposition because it handles heterogeneity in each mode, learns the clustering patterns across different modes of data in a more independent way, and provides flexibility for clustering a certain mode of the tensor without being affected by correlation with other modes. Our method is similar to those in a recent series of papers [27, 21] that use the CP decomposition structure. Note that their estimation algorithms use the framework of tensor power method [1]. In contrast, our algorithm utilizes Alternating Least Square (ALS), a widely employed decomposition algorithm [14]. The ALS algorithm estimates the CP decomposition by specifying factor matrices simultaneously, instead of the column-by-column recovery for each factor matrices implemented in tensor power methods. The ALS algorithm has been proved to be robust and computational efficient [10, 12, 15]. In particular, compared to Wang et al. [27] which performs clustering over estimated factor matrices, our method encourages smoothness in the factor matrices by imposing a generalized LASSO penalty, resulting in apparent clustering structures in factor matrices since data from same cluster will possess the same value by choosing appropriate tuning parameters. In addition, one critical assumption in Sun and Li [21] is that samples from the same cluster are ordered consecutively, an assumption that is rarely valid in applications. Our approach employs a generalized LASSO regularization that is based on a graphical distance intuition learned from initial observations. We demonstrate that the generalized LASSO penalty achieves better clustering performance when clustering assignments are randomly distributed. Even though Fused-Orth-ALS is motivated by the Orth-ALS algorithm [19], we make several advances. First, Orth-ALS only considers tensor decomposition performance under the noiseless situation, and we successfully provide statistical guarantees for Fused-Orth-ALS under any error model, with a mild constraint on error tensor spectral norm. Second, theoretical properties for Orth-ALS are established only for symmetric tensors, while Fused-Orth-ALS can achieve both recovery and clustering consistency for any asymmetric tensors.

**Notations.** Calligraphic letters such as $\mathcal{Y} \in \mathbb{R}^{d_1 \times d_2 \times d_3}$ are used to denote order three $(d_1, d_2, d_3)$-dimensional tensors. By convention, we use a colon to indicate all elements of a mode. Thus, mode-1, mode-2, mode-3 slices of $\mathcal{Y}$ can be represented as $\mathcal{Y}_{i::}, \mathcal{Y}_{:j:}, \mathcal{Y}_{::k}$. Bold uppercase and lowercase letters are used to represent matrices and vectors respectively, such as $\mathbf{A} \in \mathbb{R}^{d_1 \times d_2}$ and $\mathbf{u} \in \mathbb{R}^{d_1} = [u_1, ..., u_{d_1}]^\top$. The $i$th row and $j$th column of $\mathbf{A}$ are denoted as $\mathbf{A}_{i:}$ and $\mathbf{A}_{:j}$. For notational convenience, we define the multilinear combination of tensor with three vectors $\mathbf{u}_1, \mathbf{u}_2, \mathbf{u}_3$ as $\mathcal{Y}(\mathbf{u}_1, \mathbf{u}_2, \mathbf{u}_3) = \sum_{i,j,k} u_{1,i} u_{2,j} u_{3,k} \mathcal{Y}_{ijk}$. In particular, the spectral norm of an order three tensor $\mathcal{Y}$ is defined by $\|\mathcal{Y}\| = \max_{\|\mathbf{u}_1\|_2 = \|\mathbf{u}_2\|_2 = \|\mathbf{u}_3\|_2 = 1} \sum_{i,j,k} u_{1,i} u_{2,j} u_{3,k} \mathcal{Y}_{ijk}$. The Frobenious norm of $\mathcal{Y}$ is the natural generalization of the Frobenius norm of a matrix, $\|\mathcal{Y}\|_F = \sqrt{\sum_{i,j,k} \mathcal{Y}_{ijk}^2}$. Lastly, $[K] := \{1, 2, ..., K\}$ stands for the whole index set and we use $\circ, \odot$ to represent outer product and Khatri–Rao product. For two positive sequences $\{a_n\}, \{b_n\}$, $a_n \lesssim b_n$ means $a_n \leq C b_n$ for some constant $C > 0$ independent of $n$.

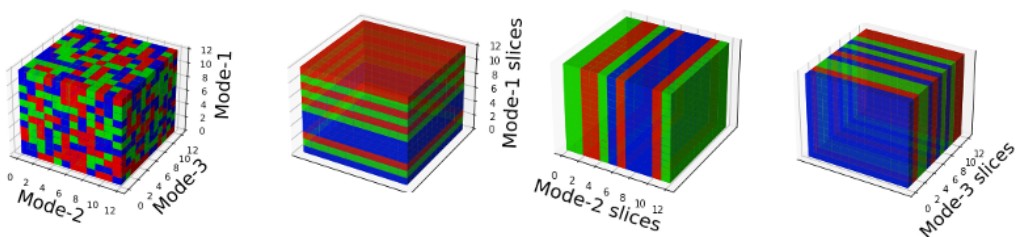

Figure 1: Fused-Orth-ALS algorithm performs multi-modes clustering through detecting cluster structures of slices along each mode.

## 2 Clustering model via tensor decomposition with regularization

Given observed tensor $\mathcal{Y} \in \mathbb{R}^{d_1 \times d_2 \times d_3}$ that is a noisy version of a tensor of interest $\mathcal{Y}^*$, i.e. $\mathcal{Y} = \mathcal{Y}^* + \mathcal{E}$, our goal is to uncover the clustering structures for the three modes of $\mathcal{Y}^*$. We assume that $\mathcal{Y}^*$ is a tensor with a rank $K$ CP decomposition structure $\mathcal{Y}^* = \sum_{i \in [K]} w_i \mathbf{A}_{:i} \circ \mathbf{B}_{:i} \circ \mathbf{C}_{:i}$, where $\mathbf{A} \in \mathbb{R}^{d_1 \times K}, \mathbf{B} \in \mathbb{R}^{d_2 \times K}, \mathbf{C} \in \mathbb{R}^{d_3 \times K}$ are factor matrices with columnwise unit norm, i.e. $\|\mathbf{A}_{:i}\|_2 = \|\mathbf{B}_{:i}\|_2 = \|\mathbf{C}_{:i}\|_2 = 1, \forall i \in [K]$, and $\mathbf{w} = [w_1, ..., w_K]^\top \in \mathbb{R}^K$ captures the weights of the factor matrices. The construction of $\mathcal{Y}^*$ leads to dependence of clustering structures on the factor matrices along each mode. Specifically, suppose there are $s_1$ clusters for the first mode, and each row in $\mathbf{A}$ can be expressed as $\mathbf{A}_{i:} = \sum_{j=1}^{s_1} \boldsymbol{\mu}_{1,j}^\top \mathbf{1}_{i \in \mathfrak{A}_j^*}$, where $\boldsymbol{\mu}_{1,j} = [\mu_{(1,j),1}, ..., \mu_{(1,j),K}]^\top$ is the mean value for rows in $j$th cluster and $\mathfrak{A}_j^*$ contains all the rows indices that belong to $j$th cluster over first mode. Consequently, the true cluster means $\bar{\mathbf{A}}_1, ..., \bar{\mathbf{A}}_{s_1}$ for slices along first mode $\{\mathcal{Y}_{1::}, ..., \mathcal{Y}_{d_1::}\}$, as indicated in Figure 1, can be written as

$$\bar{\mathbf{A}}_1 = \sum_{i=1}^{K} w_i \mu_{(1,1),i} \mathbf{B}_{:i} \circ \mathbf{C}_{:i}, \quad ..., \quad \bar{\mathbf{A}}_{s_1} = \sum_{i=1}^{K} w_i \mu_{(1,s_1),i} \mathbf{B}_{:i} \circ \mathbf{C}_{:i}$$

This reveals the core idea of clustering via CP decomposition: all the true cluster means share the same $K$ rank-1 matrix basis $\mathbf{B}_{:i} \circ \mathbf{C}_{:i}, \forall i \in [K]$. Thus, for every $i$, we would expect there exists $s \in [s_1]$ such that $i \in \mathfrak{A}_s^*$ and as a result, the clustering label for $i$th element along the first mode will be $s$. As shown in Sun and Li [21], the tensor Gaussian mixture model can be viewed as a special case of this clustering structure. However, finding the partitions $\mathfrak{A}_i^*, \forall i \in [s_1]$ is a combinatorial hard problem and could be computational intractable. Inspired by regularization term proposed in Chi et al. [8], we impose a full pairwise difference operator over rows of $\mathbf{A}$ to yield a weighted penalty on local differences,

$$\| {}^1\boldsymbol{\Delta}\mathbf{A}\|_1 = \sum_{(i_1,i_2) \in \mathcal{S}} \gamma^1_{i_1,i_2} \|\mathbf{A}_{i_1:} - \mathbf{A}_{i_2:}\|_1$$

where $\mathcal{S} := \{(i_1, i_2)| \ i_1 < i_2, i_1 \in [d_1], i_2 \in [d_2]\}$ is a set containing pairs of different row indices and parameters $\boldsymbol{\gamma}^1 = \{\gamma^1_{i_1,i_2}| \ (i_1, i_2) \in \mathcal{S}\}$ are non-negative weights that are critical for controlling the penalty imposed on pairwise row differences for $\mathbf{A}$; we will explain in detail how to choose them later. Analogously, $\boldsymbol{\mu}_{2,j}, \mathfrak{B}_j^*, {}^2\boldsymbol{\Delta}, \boldsymbol{\gamma}^2$ and $\boldsymbol{\mu}_{3,j}, \mathfrak{C}_j^*, {}^3\boldsymbol{\Delta}, \boldsymbol{\gamma}^3$ can be defined to characterize the clustering structure and regularization along the second and third modes.

In summary, we propose the following penalized constrained optimization as an approach to reveal the latent clustering structure,

$$\min_{\mathbf{A},\mathbf{B},\mathbf{C},\mathbf{w}} \|\mathcal{Y} - \sum_{i \in [K]} w_i \mathbf{A}_{:i} \circ \mathbf{B}_{:i} \circ \mathbf{C}_{:i}\|_F^2 + \lambda \Big[\| {}^1\boldsymbol{\Delta}\mathbf{A}\|_1 + \| {}^2\boldsymbol{\Delta}\mathbf{B}\|_1 + \| {}^3\boldsymbol{\Delta}\mathbf{C}\|_1\Big]$$

$$\text{s.t. } \|\mathbf{A}_{:i}\|_2 = \|\mathbf{B}_{:i}\|_2 = \|\mathbf{C}_{:i}\|_2 = 1, \forall i \in [K] \tag{1}$$

In particular, as $\lambda$ increases, $\mathbf{A}_{i:}$ will shrink towards each other, indicating pairwise differences of rows in $\mathbf{A}$ will become increasingly sparse. Sparsity in pairwise row differences naturally leads to a partitioning assignment $\mathfrak{A}_j^*, \forall j \in [s_1]$. For simplicity, we choose $\lambda$ to be the same over three modes. We would like to draw the attention to the difference between the regularization term in (1) and that in Chi et al. [8]. We use an $\ell_1$ norm penalty which will yield pairwise difference equal to 0 due to feature selection for $\ell_1$ regularization, while Chi et al. [8] adopts the Frobenious norm which only shrinks the pairwise difference to 0 gradually. As mentioned, dynamic tensor clustering [21] used a fused LASSO penalty. However, this simple fusion structure only works if samples from same cluster have consecutive indices. Obviously, our weighted pairwise difference operator ${}^i\boldsymbol{\Delta}$ is a generalization, and later we will show how our method outperforms dynamic tensor clustering when the cluster labels are randomly assigned.

## 3 Implementation: fused orthogonal alternating least squares algorithm

### 3.1 Algorithm

Starting from the alternating least square (ALS) algorithm [14], we propose the fused orthogonal alternating least squares (Fused-Orth-ALS) method described in Algorithm 1 by adding two additional

steps that are designed for multi-mode clustering. First, the orthogonalization step is performed before each iteration of ALS, which allows for the avoidance of local optima and more rapid convergence to the true factors. Moreover, a 'Fuse' operator with generalized LASSO regularization $\boldsymbol{\Delta}$ is employed on each column of ALS estimates which in general is defined as

$$\text{Fuse}(\mathbf{u}, \boldsymbol{\Delta}, \lambda) = \arg\min_{\mathbf{v} \in \mathbb{R}^d} \{\frac{1}{2}\sum_{j=1}^{d}(v_j - u_j)^2 + \lambda\|\boldsymbol{\Delta}\mathbf{v}\|_1\}$$

---

**Algorithm 1:** Fused-Orth-ALS Algorithm

---

**Input:** Tensor $\mathcal{Y}$, tensor CP rank $K$
**Output:** $\hat{\mathcal{Y}} = \sum_{i=1}^{K} \hat{w}_i \hat{\mathbf{A}}_{:i}^t \circ \hat{\mathbf{B}}_{:i}^t \circ \hat{\mathbf{C}}_{:i}^t$
Initialize $\hat{\mathbf{A}}^0, \hat{\mathbf{B}}^0, \hat{\mathbf{C}}^0$ with columns randomly from the unit sphere and iteration index $t = 0$;
**while** *termination condition is not satisfied* **do**
$\quad$ $t \leftarrow t + 1$;
$\quad$ Find QR decomposition of $\hat{\mathbf{A}}^{t-1}$, set $\hat{\mathbf{A}}^{t-1} = Q$. Orthogonalize $\hat{\mathbf{B}}^{t-1}$ and $\hat{\mathbf{C}}^{t-1}$ similarly;
$\quad$ $\mathbf{X}^t \leftarrow \mathcal{Y}_{(1)}(\hat{\mathbf{C}}^{t-1} \odot \hat{\mathbf{B}}^{t-1})$, $\tilde{\mathbf{X}}^t := [\tilde{\mathbf{X}}_{:i}^t]_{i=1}^K$ with columns $\tilde{\mathbf{X}}_{:i}^t \leftarrow \text{Fuse}(\mathbf{X}_{:i}^t, {}^1\boldsymbol{\Delta}, \lambda)$;
$\quad$ $\mathbf{Y}^t \leftarrow \mathcal{Y}_{(2)}(\hat{\mathbf{C}}^{t-1} \odot \hat{\mathbf{A}}^{t-1})$, $\tilde{\mathbf{Y}}^t := [\tilde{\mathbf{Y}}_{:i}^t]_{i=1}^K$ with columns $\tilde{\mathbf{Y}}_{:i}^t \leftarrow \text{Fuse}(\mathbf{Y}_{:i}^t, {}^2\boldsymbol{\Delta}, \lambda)$;
$\quad$ $\mathbf{Z}^t \leftarrow \mathcal{Y}_{(3)}(\hat{\mathbf{B}}^{t-1} \odot \hat{\mathbf{A}}^{t-1})$, $\tilde{\mathbf{Z}}^t := [\tilde{\mathbf{Z}}_{:i}^t]_{i=1}^K$, with columns $\tilde{\mathbf{Z}}_{:i}^t \leftarrow \text{Fuse}(\mathbf{Z}_{:i}^t, {}^3\boldsymbol{\Delta}, \lambda)$;
$\quad$ Normalize $\tilde{\mathbf{X}}^t, \tilde{\mathbf{Y}}^t, \tilde{\mathbf{Z}}^t$ with columnwise norm unit 1 and store the results as $\hat{\mathbf{A}}^t, \hat{\mathbf{B}}^t, \hat{\mathbf{C}}^t$;
**end**
Estimate weights $\hat{w}_i = \mathcal{Y}(\hat{\mathbf{A}}_{:i}^t, \hat{\mathbf{B}}_{:i}^t, \hat{\mathbf{C}}_{:i}^t), \forall i \in [K]$

---

We use the 'Fuse' operator since the regularization term in (1) can be viewed as the sum of 'Fuse' regularizations on each column of factor matrices, e.g., $\| {}^1\boldsymbol{\Delta}\mathbf{A}\|_1 = \sum_{i=1}^K \| {}^1\boldsymbol{\Delta}\mathbf{A}_{:i}\|_1$, and it can be solved efficiently via existing methods [2, 29]. After obtaining estimates $\hat{\mathbf{A}}, \hat{\mathbf{B}}, \hat{\mathbf{C}}$, clustering algorithm such as $k$-means or hierarchical clustering are used to obtain cluster assignment $\hat{\mathfrak{A}}_i, \hat{\mathfrak{B}}_j, \hat{\mathfrak{C}}_k, \forall i \in [s_1], j \in [s_2], k \in [s_3]$.

From a computational complexity perspective, orthogonalization, ALS updates, and Fuse operations in Algorithm 1 take $O(K^2(d_1 + d_2 + d_3))$, $O(Kd_1d_2d_3)$ and $O(K(d_1^3 + d_2^3 + d_3^3))$ number of operations respectively. Thus, in a general case, the total computational complexity of Fused-Orth-ALS algorithm on a $(d_1, d_2, ..., d_D)$-dimensional tensor is $O(K(\max(K\sum_{j=1}^D d_j, D\prod_{j=1}^D d_j, \sum_{j=1}^D d_j^3)))$. When $d_j$ are of the same order $d$ and tensor rank $K = o(Dd^{D-1})$, the total complexity can be simplified to $O(KDd^D)$ which is the same order as classical ALS algorithm, meaning that extra orthogonalization and 'Fuse' steps do not increase the computational complexity. The proposed clustering model is clearly distinct from dynamic tensor clustering (DTC) [21], in terms of implementation of generalized LASSO regularization, ALS updates and added orthogonalization. Our method has the same computational complexity as DTC but orthogonality leads to significant speedups in the iterations required for recovery convergence, which will be discussed in detail in section 5 later.

### 3.2 Regularization weights and tuning parameters

In practice, the choice of appropriate regularization weights $\gamma^1, \gamma^2, \gamma^3$ is critical since it affects clustering accuracy and computational efficiency. For instance, $\gamma_{i_1,i_2}^1$ characterize the shrinkage of row difference between $\mathbf{A}_{i:}$ and $\mathbf{A}_{j:}$. From a graph perspective, each row of $\mathbf{A}$ can be treated as a node in an undirected graph and nodes $i_1, i_2$ are connected by an edge with weight $\gamma_{i_1,i_2}^1$. All the edges compose the edge set $\mathcal{S}$. Clearly, large value of $\gamma_{i_1,i_2}^1$ implies high similarity between $\mathbf{A}_{i_1:}$ and $\mathbf{A}_{i_2:}$, resulting in clustering nodes $i_1, i_2$ into the same group. In particular, our adopted ${}^1\boldsymbol{\Delta}$ reduces to fused LASSO penalty utilized in DTC [21] in the case of a chain graph where $\mathcal{S} = \{\mathbf{i} = (i, i+1)| i = 1, 2, ..., d_1 - 1\}$. A well-developed strategy [20, 5, 6] is espoused for choosing regularization weights: we perform a rank-$K$ CP decomposition approximation to $\mathcal{Y}$, utilize the estimated factor matrix $\hat{\mathbf{A}}$ to quantify the 'distance' among different nodes, and then calculate the regularization weights through

$$\gamma_{i_1,i_2}^1 = \iota_{i_1,i_2}^k \exp\left(-\nu\|\hat{\mathbf{A}}_{i_1:} - \hat{\mathbf{A}}_{i_2:}\|_2^2\right) \tag{2}$$

Here, $\iota_{i_1,i_2}^k$ is an indicator function specifying whether $\hat{\mathbf{A}}_{i_2:}$ is among $k$-nearest neighbors of $\hat{\mathbf{A}}_{i_1:}$ that controls density of graph. The smallest $k$ that ensures the graph is connected is the default choice. The remaining component in (2) is a Gaussian kernel with $\nu$ as a measure of scale chosen as median Euclidean distance between the $i_1$th and $i_2$th rows that are $k$-nearest neighbors of each other. Lastly, we normalize $\sum_{(i_1,i_2)\in\mathcal{S}} \gamma_{i_1,i_2}^1 = \frac{d_1}{d_1+d_2+d_3}$ to ensure the penalty over three modes are on the same scale. Similar formulas are used for $\boldsymbol{\gamma}^2$ and $\boldsymbol{\gamma}^3$.

We choose $\lambda$ through the use of the extended Bayesian Information Criterion [21, 26, 8] by minimizing

$$\log\left(\|\mathcal{Y}-\hat{\mathcal{Y}}\|_F^2 / \prod_{i=1}^3 d_j\right) + \sum_{j=1}^3 \log d_j / \prod_{j=1}^3 d_j \times \mathrm{df}_\lambda$$

where $\mathrm{df}_\lambda$ is the number of non-zero elements in estimated factor matrices, characterizing degrees of freedom. Moreover, rank $K$ and clusters $s_1, s_2, s_3$ are usually unknown and need to be estimated from data. We use the 'elbow point' method to choose rank $K$ by plotting the recovery error, and determine $s_1, s_2, s_3$ by computing gap statistics [23].

## 4   Cluster recovery and convergence

This section provides a convergence analysis for the Fused-Orth-ALS algorithm. For simplicity, we set $d_1 = d_2 = d_3 = d$ and choose equal regularization weights over all modes. Without loss of generality, we assume $w_{\max} = w_1 \geq w_2... \geq w_K = w_{\min} > 0$.

The technical assumptions needed for our analysis are as follows:

A1: Define incoherence $\rho = \max_{i\neq j}\{|\langle\mathbf{A}_{:i},\mathbf{A}_{:j}\rangle|, |\langle\mathbf{B}_{:i},\mathbf{B}_{:j}\rangle|, |\langle\mathbf{C}_{:i},\mathbf{C}_{:j}\rangle|\} \leq \alpha/\sqrt{d}$ for some $\alpha = \mathrm{polylog}(d)$ and $K = o(d^{1/3}/\mathrm{polylog}(d)^{2/3})$. The spectral norm of $\mathcal{Y}^*$ satisfies $\|\mathcal{Y}^*\| \leq w_{\max}\alpha$.

A2: Define the initialization error as $\epsilon_0 = \max_i\{\|\hat{\mathbf{A}}_{:i}^0 - \mathbf{A}_{:i}\|_2, \|\hat{\mathbf{B}}_{:i}^0 - \mathbf{B}_{:i}\|_2\}$ which satisfies

$$\epsilon_0 \leq \min\left\{\frac{w_{\min}}{12w_{\max}} - \rho^2(K-1), \frac{w_{\min}}{72\sqrt{2}w_{\max}\alpha} - \frac{2\rho(K-1)}{\alpha}, \frac{(K-1)(\rho\xi+\xi^2)}{1-(K-1)\xi(1+\xi)}, \frac{1-(K-1)(\rho+\xi)}{(K-1)(1+\xi)}\right\}$$

with $\xi$ is a constant upper bounded by $10Kw_{\max}\alpha/(w_{\min}\sqrt{d})$.

A3: Denote the spectral norm of $\mathcal{E}$ by $\psi$ such that $\psi \leq \min\left\{w_{\min}/6, w_{\max}K/d\right\}$.

A4: Denote $M = \max_j \|\,^3\boldsymbol{\Delta}_{:j}^\dagger\|_2$ and define $\vartheta = \max_i \|\,^3\boldsymbol{\Delta}\mathbf{C}_{:i}\|_1$ that satisfies

$$\vartheta \leq \left(w_{\max}(\epsilon_0^2\alpha + 2\epsilon_0\rho(K-1) + \rho^2(K-1)) + \psi\right)/\left(2Mw_{\min}(1-\epsilon_0^2)\right)$$

Assumption A1 relaxes the requirements on the orthogonality of columns in factor matrices and imposes a weaker condition on the rank $K$ than Orth-ALS algorithm [19]. Assumption A2 specifies initialization criteria that are needed only for computational issues. The bounded perturbation is specified in A3. In particular, it can be proved with high probability if elements in $\mathcal{E}$ are i.i.d from a sub-Gaussian distribution as stated in Tomioka and Suzuki [24]. The bounded fusion assumption, A4, restricts clustering complexity on factor matrices. To illustrate its meaning, consider a simple scenario when all clusters along the third mode have the same size $d/s_3$. For $\mathbf{C}_{i:}$ belonging to the same cluster $\mathfrak{C}_j^*$, they should take the same value $\boldsymbol{\mu}_{3,j}$ under our model assumption. Consequently, A4 can be rephrased as $\vartheta = O(d^{1.5}(1-1/s_3))$; in the special case when $\mathbf{C}$ has only one cluster, e.g. $s_3 = 1$, A4 reduces to $\|\,^3\boldsymbol{\Delta}\mathbf{C}\|_1 = 0$. Thus, bounding $\vartheta$ is equivalent to bounding cluster size $s_3$ when sample size over third mode is kept fixed.

We can now derive a recovery error bound for the Fused-Orth-ALS algorithm.

**Theorem 1** *Assuming assumptions A1-A4 hold, factor matrix estimate $\hat{\mathbf{C}}_{:i}, \forall i \in [K]$ of Algorithm 1 satisfies the following error bound with high probability*

$$\|\hat{\mathbf{C}}_{:i} - \mathbf{C}_{:i}\|_2 \lesssim \frac{w_{\max}\rho^2(K-1)}{w_{\min}} + \psi/w_{\min}$$

*when choosing appropriate $\lambda$. Similar error bounds hold for the other two factor matrices estimates $\hat{\mathbf{A}}, \hat{\mathbf{B}}$.*

The error bound derived in Theorem 1 reveals how the weight $w_i$, incoherence parameter $\rho$, rank $K$ and perturbation level $\psi$ interact with each other on affecting the convergence behavior of Fused-Orth-ALS algorithm. Lower perturbation level $\psi$, lower incoherence parameter $\rho$, rank $K$ and lower signal ratio of $w_{\max}/w_{\min}$ all result in lower error bounds. Clearly, the error bound for each column in factor matrices has two parts: one is related to perturbation level $\psi$, the other is dependent on the underlying CP decomposition structure, signal ratio $w_{\max}/w_{\min}$, rank $K$ and dimension $d$ along each mode. Thus, under high dimensional settings when $d \to \infty$, i.e., $\rho^2 \le \alpha^2/d \to 0$, the second part will dominate the error bound while if tensor data is almost noiseless, e.g. $\psi \approx 0$, the first part will be the main source of error. The following corollary analyzes the relationship between these two parts under a special case when the error tensor follows a sub-Gaussian distribution.

**Corollary 1** *Assume assumptions A1-A4 hold. If we further assume each element in error tensor, $\mathcal{E}_{ijk}$, is independent, with zero mean and satisfies $\mathbb{E}[e^{t\mathcal{E}_{ijk}}] \le e^{\frac{\sigma^2 t^2}{2}}$, $w_{\max}/w_{\min} \le C$ where $C$ is positive constant and the minimal weight satisfies*

$$w_{\min} \succ \sqrt{\sigma^2 \Big[ 3d \log \frac{6}{\log 3/2} + \log \frac{2}{\delta} \Big] d^2/(K-1)^2}$$

*then $\hat{\mathbf{C}}_{:i}$, $\forall i \in [K]$ from Algorithm 1 satisfies the following error bound,*

$$\|\hat{\mathbf{C}}_{:i} - \mathbf{C}_{:i}\|_2 \lesssim (K-1)/d$$

*with probability at least $1 - \delta$. Same error bounds hold for the other two factor estimates $\hat{\mathbf{A}}, \hat{\mathbf{B}}$.*

Thus, by imposing the sub-Gaussian distribution assumption on error tensor elements, recovery consistency for factor matrices estimated from Fused-Orth-ALS algorithm can be established.

Next, we demonstrate cluster consistency for the clustering algorithm performed on factor matrices recovered by Fused-Orth-ALS algorithm. Since clustering algorithm is performed on rows of $\hat{\mathbf{A}}, \hat{\mathbf{B}}, \hat{\mathbf{C}}$, clustering error is quantified through the true mean value and its estimate $\|\boldsymbol{\mu}_{1,j_1} - \hat{\boldsymbol{\mu}}_{1,j_1}\|_2, \|\boldsymbol{\mu}_{2,j_2} - \hat{\boldsymbol{\mu}}_{2,j_2}\|_2, \|\boldsymbol{\mu}_{3,j_3} - \hat{\boldsymbol{\mu}}_{3,j_3}\|_2$.

**Theorem 2** *Assume assumptions in Corollary 1 hold, we have*

$$\max_{j_1}\|\hat{\boldsymbol{\mu}}_{1,j_1} - \boldsymbol{\mu}_{1,j_1}\|_2 \lesssim K^{1.5}/d$$

$$\max_{j_2}\|\hat{\boldsymbol{\mu}}_{2,j_2} - \boldsymbol{\mu}_{2,j_2}\|_2 \lesssim K^{1.5}/d$$

$$\max_{j_3}\|\hat{\boldsymbol{\mu}}_{3,j_3} - \boldsymbol{\mu}_{3,j_3}\|_2 \lesssim K^{1.5}/d$$

*hold with probability at least $1 - \delta$. Furthermore, if $\min_{j_1 \in \mathfrak{A}_m^*, j_1' \in \mathfrak{A}_{m'}^*, m \ne m'} \|\boldsymbol{\mu}_{1,j_1} - \boldsymbol{\mu}_{1,j_1'}\|_2 \gtrsim K^{1.5}/d$, $\min_{j_2 \in \mathfrak{B}_n^*, j_2' \in \mathfrak{B}_{n'}^*, n \ne n'} \|\boldsymbol{\mu}_{2,j_2} - \boldsymbol{\mu}_{2,j_2'}\|_2 \gtrsim K^{1.5}/d$, $\min_{j_3 \in \mathfrak{C}_l^*, j_3' \in \mathfrak{C}_{l'}^*, l \ne l'} \|\boldsymbol{\mu}_{3,j_3} - \boldsymbol{\mu}_{3,j_3'}\|_2 \gtrsim K^{1.5}/d$, we have $\hat{\mathfrak{A}}_m = \mathfrak{A}_m^*$, $\hat{\mathfrak{B}}_n = \mathfrak{B}_n^*$, $\hat{\mathfrak{C}}_l = \mathfrak{C}_l^*$ hold with probability at least $1 - \delta$.*

This theorem shows that clustering consistency holds as long as $K^{1.5}/d \to 0$, which allows rank $K$ increase with the dimension $d$. At first look, the conclusion in Theorem 2 seems to indicate that the cluster mean error bound does not depend on number of clusters over each mode, $s_1, s_2, s_3$. However, as we stated before, assumption A4 employs an "invisible" bound on the number of clusters which is closely related with the convergence rate derived in Theorem 2. We provide a detailed explanation in supplementary material stating cluster mean error bound $K^{1.5}/d$ increases with the number of clusters $s_i$. Note that Theorem 2 assumes that true rank $K$ is known; we leave the effect of an estimated rank $\hat{K}$ on clustering consistency for further study.

## 5 Numerical experiments

### 5.1 Synthetic datasets

In this section, we investigate the performance of the Fused-Orth-ALS algorithm on small samples and compare the recovery and clustering error with alternative tensor-based clustering methods.

Recovery error is evaluated through $\|\hat{\mathcal{Y}} - \mathcal{Y}^*\|_F/\|\mathcal{Y}^*\|_F$. Clustering error measures the frequency of mismatches between the estimated clustering assignment $\hat{\mathfrak{M}}$ and the true $\mathfrak{M}$ over samples $x_1, ..., x_n$, which is defined as

$$\left|\{(i,j) : \mathbf{1}_{\hat{\mathfrak{M}}(x_{j_1})=\hat{\mathfrak{M}}(x_{j_2})} \neq \mathbf{1}_{\mathfrak{M}(x_{j_1})=\mathfrak{M}(x_{j_2})}, j_1 < j_2, j_1, j_2 \in [n]\}\right|/\left[n(n-1)/2\right]$$

and can be easily computed after obtaining the cluster assignment $\hat{\mathfrak{A}}_m, \hat{\mathfrak{B}}_n, \hat{\mathfrak{C}}_l, \forall m \in [s_1], n \in [s_2], l \in [s_3]$. Note that the clustering error is similar to two other commonly used metrics: adjusted Rand index [11] and variation of information [17]. For the simulations below, elements in the error tensor $\mathcal{E}$ are generated independently from a Gaussian distribution with mean 0 and variance $\sigma^2$. We compare Fused-Orth-ALS Algorithm with three methods: dynamic tensor clustering (DTC) [21], CP-Kmeans algorithm which performs a rank-$K$ CP decomposition on the tensor observations first and then independently applies $k$-means algorithm clustering to the rows of derived factor matrices, and multiway clustering for tensor block models (TBM)[26].

### 5.1.1 Single-mode clustering

We first generate an order three tensor with CP-rank 2 for which we would like to do single-mode clustering along the third mode. The dimension of the first and second modes are set the same ($d_1 = d_2 = d$) and their unnormalized columns are

$$\mathbf{A}_{:1} = \mathbf{B}_{:1} = (\mu, -\mu, 0.5\mu, -0.5\mu, \underbrace{0, ..., 0}_{d-4})^\top, \mathbf{A}_{:2} = \mathbf{B}_{:2} = (0, 0, 0, 0, \mu, -\mu, 0.5\mu, -0.5\mu, \underbrace{0, ..., 0}_{d-8})^\top$$

The third factor matrix with unnormalized & unshuffled columns is generated using

$$\mathbf{C}_{:1} = (\underbrace{\mu, ..., \mu}_{\lfloor d_3/2 \rfloor}, \underbrace{-\mu, ..., -\mu}_{\lfloor d_3/2 \rfloor})^\top, \mathbf{C}_{:2} = (\underbrace{-\mu, ..., -\mu}_{\lfloor d_3/4 \rfloor}, \underbrace{\mu, ..., \mu}_{\lfloor d_3/2 \rfloor}, \underbrace{-\mu, ..., -\mu}_{\lfloor d_3/4 \rfloor})^\top \qquad (3)$$

Then, the rows of $\mathbf{C}$ are shuffled randomly. There are four clusters over the third mode with cluster means $(\mu, -\mu), (\mu, \mu), (-\mu, \mu), (-\mu, -\mu)$ respectively. After normalizing the columns of $\mathbf{A}, \mathbf{B}, \mathbf{C}$, we calculate the weights $w_i$. Even though this example is inspired from Sun and Li [21], shuffling the rows of $\mathbf{C}$ leads to additional clustering challenges since a more sophisticated regularization is required to capture the clustering features. $\mu$ reflects cluster difficulty as it determines the distance between different clusters. A smaller $\mu$ leads to a more challenging clustering problem. For simplicity, the factor matrices used in this example, $\mathbf{A}, \mathbf{B}, \mathbf{C}$, are orthogonal, satisfying assumption A1.

We set $d = 20, d_3 = 48, \mu = 1$, vary $\sigma \in \{0, 0.25, 0.5, 0.75, 1\}$ and compare the performance of the four algorithms by reporting recovery and clustering errors. As summarized in Figure 2 (a) and (b), Fused-Orth-ALS outperforms the other three methods under different values of perturbation, achieving the best recovery error and clustering error. Not surprisingly, TBM performs worst due to model misspecification: the formulation of simulated factor matrices leads to an underlying cluster structure for mode three instead of a multiway block structure for all three modes. Figure 2 (c) and (d) show that average running times are comparable for Fused-Orth-ALS and DTC where the orthogonalization step added in Fused-Orth-ALS still decrease number of iterations required for convergence. Furthermore, to illustrate the advantage of generalized LASSO regularization imposed on Fused-Orth-ALS over fused LASSO penalty adopted in DTC, we vary the value for $\mu \in \{0.1, 0.7\}$ to mimic different levels of clustering complexity and set $d = 8, \sigma = 0.001, d_3 = 20$. It is obvious from Table 1 that our method outperforms DTC especially as the 'signal' $\mu$ becomes weak.

Table 1: Comparison of Fused-Orth-ALS and DTC under different signal levels. Average errors and standard deviations (in parenthesis) are reported based on 50 replications.

| | $\mu = 0.1$ | | $\mu = 0.7$ | |
|---|---|---|---|---|
| | Fused-Orth-ALS | DTC | Fused-Orth-ALS | DTC |
| Recovery error | **0.8198(0.2034)** | 1.0538(0.0224) | **0.0011(0.0001)** | 0.0069(0.0006) |
| Clustering error | **0.1902(0.1475)** | 0.3847(0.0394) | **0(0)** | 0(0) |

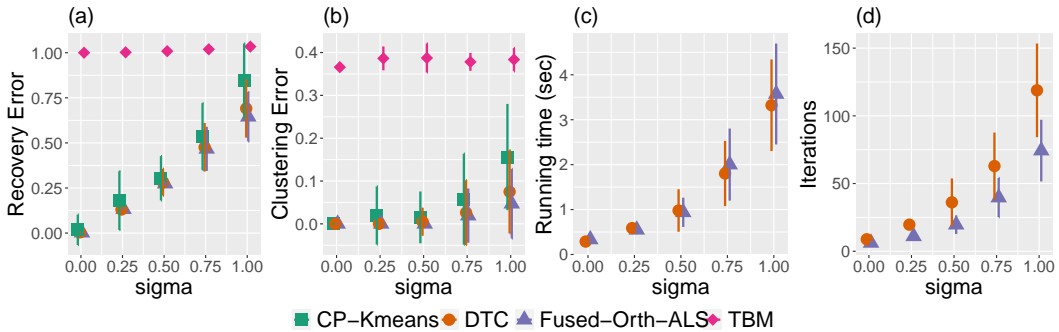

Figure 2: Comparison of single-mode clustering for Fused-Orth-ALS, DTC CP-Kmeans and TBM

### 5.1.2 Multi-mode clustering

To validate the performance on multi-mode clustering, we extend the simulation settings above by imposing clustering structures on all three factor matrices $\mathbf{A}, \mathbf{B}, \mathbf{C}$, similarly to (3). The corresponding choice of cluster means are: $(\mu, 0.5\mu), (-0.5\mu, \mu), (0, -\mu), (-\mu, 0)$ for mode 1, $(0, \mu), (-\mu, 0), (0.1\mu, -\mu), (-\mu, -0.1\mu)$ for mode 2 and $(\mu, \mu), (-\mu, \mu), (\mu, -\mu), (-\mu, -\mu)$ for mode 3. Accordingly, each mode has four clusters, i.e., $s_1 = s_2 = s_3 = 4$ of same size: $\lfloor d_1/4 \rfloor, \lfloor d_2/4 \rfloor, \lfloor d_3/4 \rfloor$ respectively. For example, for the first mode, there are $\lfloor d_1/4 \rfloor$ rows of the factor matrix taking value $(\mu, 0.5\mu)$, $\lfloor d_1/4 \rfloor$ rows taking value $(-0.5\mu, \mu)$, $\lfloor d_1/4 \rfloor$ rows taking value $(0, -\mu)$ and the rest rows taking value $(\mu, 0)$. We fix $d_1 = d_2 = 20, d_3 = 40, \mu = 1$ and vary $\sigma \in \{1, 2, 3, 4, 5\}$. Since the detailed comparison between Fused-Orth-ALS and DTC has been illustrated above, we omit DTC (originally designed for dynamic single mode clustering of tensors). Comparison results can be found in Figure 3. It is clear that Fused-Orth-ALS algorithm outperforms the other two methods, especially under the noisy case when $\sigma$ is large.

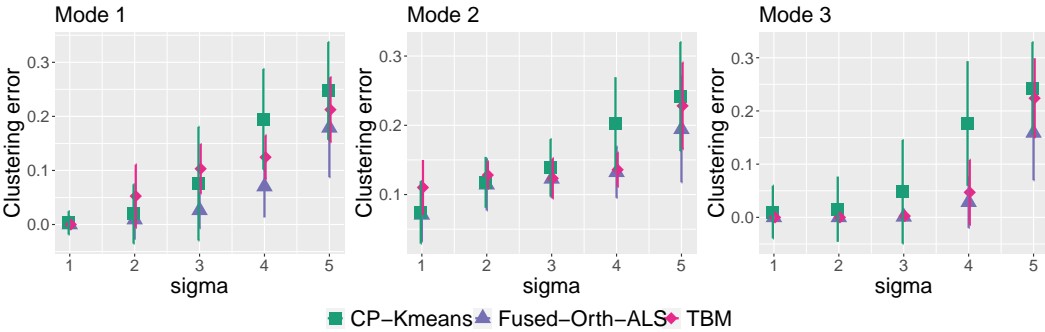

Figure 3: Comparison of multi-modes clustering for Fused-Orth-ALS, CP-Kmeans and TBM

### 5.2 Real datasets

Experiments are conducted on two real datasets: the brain node structural connectivity from Human Connectome Project (HCP) [1] [25] and political relationships between nations [2] [13]. Details for choosing rank $K$ and number of clusters are provided in supplementary material.

The HCP dataset is formatted as $\mathcal{Y} \in \mathbb{R}^{68 \times 68 \times 136}$, and consists of brain connectivity among 68 brain nodes for 136 individuals. Each entry takes on ordinal value $\{0, 1, 2\}$ which indicates the strength level of connectivity $\{$low, moderate, high$\}$ between different brain nodes. Clustering results in Table 2 capture spatial connectivity between hemispheres of brain (the first character in the node name indicates the left or right hemisphere and the number in the parenthesis indicates the node count with same name). Cluster I and II mainly capture connectivity in either left or right hemisphere while

---

[1] Dataset is available at `http://www.humanconnectomeproject.org/`

[2] Dataset is available at `http://www.charleskemp.com/code/irm.html`.

cluster III represents the cross-section connection between left and right hemisphere. *r.supramarginal* and *l.supramarginal* are picked as smaller clusters - those regions are known to play a critical role in visual word recognition and reading.

Table 2: Clustering result for 68 brain nodes in HCP dataset

| Cluster | Brain Nodes |
|---------|-------------|
| I | l.insula,l.superiortemporal(3),l.middletemporal(3) l.inferiortemporal(3),l.inferiorparietal,l.lateraloccipital(2) |
| II | r.insula,r.superiortemporal(3),r.middletemporal(3), r.inferiortemporal(3),r.lateraloccipital(2),r.precuneus,r.lingual |
| III | l.superiorfrontal(3),l.frontalpole ,l.caudalmiddlefrontal, l.parstriangularis, l.parsopercularis,l.precentral,l.temporalpole,l.postcentral, l.superiorparietal,l.medialorbitofrontal,l.isthmuscingulate,l.precuneus, l.cuneus,l.parahippocampal,l.lingual, r.superiorfrontal(3),r.frontalpole,r.caudalmiddlefrontal,r.parstriangularis, r.parsopercularis,r.precentral,r.temporalpole,,r.postcentral, r.superiorparietal,r.inferiorparietal,r.medialorbitofrontal,r.isthmuscingulate, r.cuneus,r.parahippocampal |
| IV | r.supramarginal(4) |
| V | l.supramarginal(4) |

The Nations dataset is formatted as $\mathcal{Y} \in \mathbb{R}^{14 \times 14 \times 56}$ consisting of 56 political relationships of 14 countries between 1950-1965. Each entry represents the presence or absence of a political action, such as 'treaties', 'send tourists to ' between different nations. Since 78.9% entries are zero in this dataset, we include an $\ell_0$ penalized tensor block model (denoted as 'TBM-Sparse') for comparison. The clustering result in Table 3 for Fused-Orth-ALS algorithm assigned the nations into three clusters, one representing western-bloc countries (Cluster I), one communist bloc (Cluster III), one neutral bloc (Cluster II). This result is consistent with the structure of political environment after world war II. In addition, the goodness-of-fit (proportion of variance explained) for five methods is provided in Table 4. For the nations dataset, Fused-Orth-ALS algorithm shows the highest variance explained.

Table 3: Clustering result for 14 nations in Nations dataset

| Cluster | Characteristic | Country |
|---------|----------------|---------|
| Cluster I | Western | Brazil, Netherlands, UK, USA |
| Cluster II | Neutral | Burma, Egypt, Israel, Jordan,India, Indonesia |
| Cluster III | Communist | China, Cuba, Poland, USSR |

Table 4: Comparison of goodness-of-fit for HCP and nations dataset

|  | Fused-Orth-ALS | DTC | TBM | TBM-Sparse | CP-Kmeans |
|--|----------------|-----|-----|------------|-----------|
| HCP | 0.921 | 0.921 | 0.925 | - | 0.921 |
| Nations | 0.522 | 0.458 | 0.439 | 0.433 | 0.324 |

## 6   Conclusions

This paper studies the properties and convergence rates for a novel multiway tensor clustering algorithm, Fused-Orth-ALS. The underlying CP decomposition structure imposes low-rank constraints on the tensor observation, and graph similarity based fusion regularization encourages smoothness on factor matrices, thus leading to automatic clustering. The Fused-Orth-ALS algorithm can handle sparse and dense data tensors, and achieves clustering consistency even under model misspecification. An open problem is developing theory for rank selection consistency. Another interesting topic is on clustering consistency if there are missing entries in the data. We leave these directions for further study. There are no foreseeable negative social impacts of this work.

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
