# OpenReview forum: "Fused Orthogonal Alternating Least Squares for Tensor Clustering"
_NeurIPS.cc/2022/Conference — NeurIPS 2022 Accept_

### Official Review · Reviewer_c3Lt · 2022-07-09

**Rating:** 6
**Confidence:** 2
**Soundness:** 3 good
**Presentation:** 3 good
**Contribution:** 3 good

**Summary:**

This paper proposes a new formulation and a new ALS based algorithm for tensor clustering. Upper bound of the theoretical convergence rate of this algorithm is provided, and experimental results show that this algorithm yields more accurate clustering for multiple cases compared to existing methods.

**Questions:**


Line 70: spectrum norm formulation is wrong, I guess the authors mean max rather than min.

Line 79: There is one redundant ) at the end of this line.

**Limitations:**

There are no negative social impacts of this work.

**Strengths And Weaknesses:**

Strengths:

- The problem of tensor clustering is important
- Detailed theoretical analysis of the algorithm, and better experimental results compared to existing work.

Weaknesses:

- As is declared in the paper: "First, the orthogonalization step is performed before each iteration of ALS, which allows for the avoidance of local optima and more rapid convergence to the true factors". There is no detailed analysis in the paper to show the advantage of this orthogonalization step. Additional experiments that compare the algorithm with orthogonalization and the one without orthogonalization would be helpful.
- The formulation (1) includes multiple hyperparameters. The choice of these hyperparameters in the paper is presented in 3.2 However, questions still remain as to 1) is the formulation robust to these hyperparameters, and 2) is there other reasonable ways to choose these hyperparameters, and discussions over them would be helpful.

-------
After rebuttal: I thank authors for the detailed response. My concerns are addressed.

---

> ### Author Response · Authors · 2022-08-02
> **Thanks for the comments and hope our answers address your concerns**
>
> We thank the reviewer for the constructive comments and supportive rating. We believe that our responses clarify the issues the reviewer raised.
>
> *On the advantages of orthogonalization:*
>
> We performed several experiments to show the quicker convergence of adding orthogonalization. Results are provided in Figure 2(d) where the experiment is based on the comparison of dynamic tensor clustering algorithm (which adopts the similar CP decomposition assumption, but the algorithm does not contain orthogonalization step) and our proposed method. It provides strong evidence that adding orthogonalization truly speeds up the convergence. We thank the reviewer’s constructive feedback and we will include additional experimental results to compare the ALS based algorithm with/without orthogonalization in the revised appendix.
>
> *On the choice of hyperparameters and robustness:*
>
> We fully agree that multiple hyperparameters make it difficult to implement the algorithm. During our experiments, we found that several techniques facilitate the algorithm to obtain more robust and accurate clustering performance, including choosing $\tau$ as median Euclidean distance between any pair of rows that are k-nearest neighbors of each other and normalizing $\gamma_{i_1, i_2}^1, \gamma_{i_1, i_2}^2, \gamma_{i_1, i_2}^3$; see lines 152-156.  We also considered other hyperparameter tuning methods such as cross-validation and stability selection. Since both methods are based on resampling, they are unattractive in the tensor setting due to the computational burden compared to extended Bayesian Information Criterion. In terms of robustness, we investigated this issue for the hyperparameter lambda but less for gamma because of the huge computational burden. We thank the reviewer for raising this problem; choosing optimal hyperparameters could be a productive direction for future work!
>
> In addition, we also thank the reviewer for pointing out a couple of typos. We apologize for missing them in our proofreading; they will be corrected in the revised manuscript.

---

### Official Review · Reviewer_cTv9 · 2022-07-10

**Rating:** 7
**Confidence:** 4
**Soundness:** 3 good
**Presentation:** 4 excellent
**Contribution:** 3 good

**Summary:**

This paper tackles the tensor multi-mode clustering problem. The authors model the tensor to cluster as the sum of two terms, the signal which admits a rank-K CANDECOMP/PARAFAC (CP) decomposition, and some noise. They establish a link between the CP decomposition factors and the means of the clusters from the corresponding modes to propose the core optimization problem described in the paper. They aim to approximate as closely as possible the tensor by a rank-K CP decomposition, with a generalized Lasso penalty that limits the number of unique rows in the CP decomposition factors. An Alternating Least Squares (ALS) strategy is used to solve the optimization problem one factor at a time. An orthogonalization step is added before each ALS step, and a Fuse operator is applied after the ALS step.
The algorithm is presented with convergence guarantees, and the proposed approach is compared with other tensor mode clustering methods on both simulated and real datasets. The authors also explain how to choose every parameter of the proposed algorithm.

**Questions:**

I start with an essential question determining my acceptance of the paper.
Please correct me if I am wrong, but it seems that the proof of Theorem 1 bites its tail. Indeed, Theorem 1 gives an upper bound on the distance between the true and the estimated factors. However, its proof is based on Lemma 3, which assumes this same upper bound. Furthermore, it is written in Lemma 3 that this assumption can be made "without loss of generality". I may be missing something here, so please enlighten me.

I will now ask some questions, give suggestions to improve readability and point out mistakes or typos.
- Could it be possible to assess the validity of the A1-A4 assumptions on a given dataset? Were they all valid on the simulated datasets?
- General remark: it would help the reader if different indices were used for the rows and the columns. For instance, use $k$ for columns and $i$ for rows: $\mathbf A_{:k}$ and $\mathbf A_{i:}$.
- Line 71: the definition of the spectral norm should be $ \Vert \mathcal Y \Vert = \text{max}_{\Vert \mathbf u_1 \Vert_2 = \Vert \mathbf u_2 \Vert_2 = \Vert \mathbf u_3 \Vert_2 = 1} \mathcal Y(\mathbf u_1, \mathbf u_2, \mathbf u_3)$.
- Line 79: "rank-K CP decomposition" and no closing parenthesis at the end of the line.
- Line 80: "unit ~~one~~ norm".
- Line 121: it should be [...] $+ \lambda \Vert \boldsymbol \Delta \mathbf u \Vert_1$. $\boldsymbol \Delta$ is also implicitly defined as specific matrices per mode were previously introduced.
- Line 169: $\gamma$ is used for the signal ratio, there is no possible confusion with the regularization parameters, but maybe another symbol should be preferred.
- Line 170: in A4, constant $M$ has not been introduced before.
- Line 172: "on **the** rank $K$".
- Line 191: observation made that large $\gamma$ results in lower error bounds seems wrong according to the bound given in Theorem 1.
- Line 200: what does the condition on $w_\text{min}$ means? Is it a high signal-to-noise ratio condition? Was it verified in the simulations?
- Lines 256-257: the motivation for adding orthogonality was, according to line 119, to get more rapid convergence, but here, a concern is addressed that this may increase the number of iterations. Talking about verification in lines 256-257 would be more logical.

- Appendix line 65, on the second line of the equation defining $\Delta$ (maybe another symbol would be a good choice not to get confused with $\boldsymbol \Delta$): why does this become an upper bound with $\mathbf A_{: i}$ replaced with $(\mathbf A_{: i} + \hat \xi_i)$?
- Appendix line 68 (and after): $II_{11}'$ is used but never introduced. It happens after with different quantities.
- Appendix line 71, on the first line of the equation: $\mathcal Y^*$ should be replaced with $w_l$.
- Appendix line 73, on the first of the equation: $w_l$ is missing.
- Appendix line 79: indices are wrong. It should be $j_1$ for the $\mathbf A$ terms and $j_2$ for the $\mathbf B$ terms. I get the idea of the majorization, but it is not well-written: the sums have been removed, but there are still terms in $j_1$ (and normally $j_2$) appearing in the upper bound.
- Appendix line 83, equation 11: the term $2 w_\text{max} \rho$ supposes that $j_1 \neq j_2$, but we could have the equality.
- Appendix line 84, the whole bound should be multiplied by $((K - 1) \Delta)^2$ and the last term is $(K - 2) \rho^2 w_\text{max}$.
- Appendix line 146: $\Vert . \Vert_2$.
- Appendix line 149 (and after): $\alpha^2$ turned into $\alpha$.
- Appendix line 155: I do not get how the upper bound is simplified.
- Appendix line 156: I do not get the bounds on $| 1 / \kappa |$, and I do not get how the upper bound is simplified.

**Limitations:**

The authors identified that their convergence guarantees were valid for the real rank $K$ and that more work is needed to derive results for an estimated rank $K$. The convergence results show that if the number of samples goes to infinity and the tensor is noiseless, the true factors will be retrieved. However, it does not say anything about the algorithm reaching a stationary point. Is it possible to show, for example, that the objective function decreases monotonously?

**Strengths And Weaknesses:**

### Strengths
1. The paper is clear and well written. The progression is logical and the choices made are well justified.

2. The proposed approach is compared favorably with concurrent methods on simulated and real datasets.

3. Theoretical convergence properties of the given algorithm are derived.

### Weaknesses
4. I have concerns about the proof of Theorem 1. See the section **Questions** below.

5. No code is given to reproduce the results.

**Update after rebuttal**: the authors have provided code (although I did not have time to go through it) and addressed my concerns regarding the proof of Theorem 1. I think that the clarity of the proof can still be improved, but the authors are working on it, and I have good hope that they will manage to do so in the given amount of time.

---

> ### Author Response · Authors · 2022-08-02
> **Thanks for the comments and hope our answers address your concerns**
>
> We thank the reviewer for the very detailed report. The reviewer raised an important issue on Theorem 1 that will determine the final score and the acceptance of paper. We clarify the issue below and hope that our answer is satisfactory.
>
> *On proof of Theorem 1 and connection to Lemma 3:*
>
> We thank the reviewer for pointing out the unclear aspects in the proof of Theorem 1 and its connection to Lemma 3. We agree with the reviewer assessment, and in fact have already (after submission) reorganized the appendix to flow more smoothly and clarify the raised issue. We think it’s much easier to read now and we are sorry that you had to struggle through the old version. In short, the Lemma 3 assumption is for the factors that have already converged and the bound is in the same form as the bound of Theorem 1 for estimated factors.
>
> The strict mathematical derivations have been provided in the revised appendix and we provide a brief explanation of the proof logic here: we mimic the proof idea of Lemma 3 in Sharan, V. and Valiant, G. (2017). We prove that Fused-Orth-ALS recovers the remaining factors by induction. The first step is the base case (which is proved in the first half part of appendix A) and the algorithm recovers the first factor. We next show that if the first $(m−1)$ factors have converged, then the $m$th factor converges with high probability. The main idea is that as the factors have small correlation with each other, hence orthogonalization does not affect the factors which have not been recovered but ensures that the $m$th estimate never has high correlation with the factors which have already been recovered. The key idea for Lemma 3 is that the orthogonal basis is close to the original factors as the factors are incoherent. We have modified Lemma 3 in the updated appendix and we believe that the modified appendix addresses your concern.
>
> *On readability and typos:*
>
> The reviewer provided a lot of excellent suggestions for improving readability. We will use all of them in the revised version. Some notes on a couple of the suggestions are below:
>
> *On validity of assumptions in real and simulated datasets:* Please note that some assumptions are common in tensor decomposition papers, and some can only be validated with high probability under some special cases.
>
> - Assumption 1: incoherence condition is commonly used in the tensor decomposition literature (Anandkumar et al., 2014; Sun et al., 2017; Sun and Li, 2019) which relaxes the requirements on the orthogonality of columns in factor matrices. Anandkumar et al. (2014) provides detailed proof that it is satisfied if columns of factor matrices are uniformly and independently drawn from the unit sphere. The additional constraint on the spectral norm of factor matrices can be proved to be satisfied with high probability using the bounded tensor spectral norm (Tomioka and Suzuki (2014)).
> - Assumption 2 restricted initialization for Fused-Orth-ALS algorithm to be related with weights ratio $\gamma$, rank $K$ and dimension $d$. Please note that this assumption is only needed for computational issues; detailed explanations for each term can be found in appendix A.
> - Assumption 3 bounds the perturbation level in terms of the spectral norm of error tensor $\psi$. This can be satisfied with high probability if each element in the error tensor $\mathcal{E}$ follows an i.i.d sub-Gaussian distribution as stated in Tomioka and Suzuki (2014).
> - Assumption 4: as we explained in Line 175-181, it puts some restrictions on the clustering complexity of factor matrices, which is usually difficult to verify in a real dataset.
>
> We believe this is a direction for future work from us and from the community of researchers focusing on this topic.
>
> *On the constant $M$ in assumption 4:* We apologize for not defining $M$ before discussing A4. It is introduced in the condition on tuning parameter choice $\lambda$ in Theorem 1, where $M$ is the maximum $\ell_2$ norm of the columns of pairwise difference operator over rows of $C$. We will add this detailed explanation before introducing the assumptions in the revised manuscript.
>
> *On condition of $w_\min$:* We interpret $w_\min$ as the signal strength. Since the condition in Line 200 (Corollary 1) is a lower bound order on $w_\min$, it is hard to practically verify it. We will include additional experiments in the appendix to show how the recovery and clustering performance get worse when the signal strength $w_\min$ gets weak.
>
> *On the convergence of the algorithm and monotonicity of the objective function (in Limitations):* The reviewer raised an important aspect on the monotonicity of the objective function as the algorithm progresses. In the experiments we performed, the objective function does not decrease monotonously (not a convex function). The plots of the objective function versus iteration number shows an overall decreasing trend with small local oscillations.

---

> > ### Comment · Reviewer_cTv9 · 2022-08-05
> > **On proof of Theorem 1 and connection to Lemma 3**
> >
> > I thank the authors for their explanations and for considering my comments. I will dive deeper into the new version of the manuscript in the following days but can make the following remark so far.
> >
> > I now think that the proofs of Theorem 1 and Lemma 3 are sound, even if I need extra time to check the details carefully. However, I think the way it is written is confusing. I believe that merging Lemma 3 into Theorem 1 would make things easier. Indeed, in the proof of Lemma 3, line 162, it is stated that $\Vert \hat{\xi}_p \Vert_2 \leq 6 \sqrt{2} (\alpha^2 + 1) K \gamma / d$, however, this can only be proven using the induction hypothesis and is the main result of the proof of Theorem 1. On the other hand, in the proof of Theorem 1, in lines 66 and 75, Lemma 3 is invoked, but only the induction hypothesis is required. What do you think of this suggestion?

---

> > > ### Author Response · Authors · 2022-08-07
> > > **Thanks for your comment!**
> > >
> > > We sincerely appreciate the hard work and time of the reviewer on the revised supplementary and manuscript!
> > >
> > > Yes, the proof process of theorem 1 requires the induction result of Lemma 3 and Lemma 3 takes the result of Theorem 1 as assumption. Our original concern is incorporating the proof of Lemma 3 into Theorem 1 might give the reader a sense of confusing since we switch the gear a lot and take some effort to show the orthogonalization does not affect the factors which have not been recovered but ensures that the mth estimate never has high correlation with the factors which have already been recovered. But now it seems combining those two parts of proof (Theorem 1 and Lemma 3) makes more sense and is easier to understand. We thank your constructive comments and will modify it in the manuscript.
> > >
> > > Thanks!

---

> > > > ### Comment · Reviewer_cTv9 · 2022-08-08
> > > > **On the proof of Theorem 1**
> > > >
> > > > I have spent some more time on the proof of Theorem 1 and have further questions and concerns. I understand that two things are done at the same time in the proof: write $II_1$ as a sum of two vectors, one proportional to $w_i \mathbf C_{: i}$ and $\boldsymbol \Lambda$, and find an upper bound on the norm of $\boldsymbol \Lambda$. Let's write the first vector as $\nu_0 w_i \mathbf C_{: i}$.
> > > >
> > > > The first issue is that the first term $\nu_0 w_i \mathbf C_{: i}$ needs to be exact, but some upper bounds are used in lines 72, 77, and 78. Norms and vectors are mixed, and some inequalities do not make sense, like in lines 79 and 82 (they should be comparing norms and not vectors). Furthermore, all $\xi$ appearing in bounds should be $\Vert \xi \Vert_2$.
> > > >
> > > > The second (probably minor) issue is the upper bound of $\Vert II_{14} \Vert_2$. Equation (11) is true when $j_1 \neq j_2$, when summing, the equality case appears $(i-1)$ times and modifies the upper bound:
> > > > $\Vert II_{14} \Vert_2 \leq (K - 1) \Delta^2 w_\text{max} [(K - 1) (\xi^2 \alpha + \rho^2 + 2 \xi + 2 \rho \xi (K - 1)) + (K - 2)^2 \rho^2 + 2 \rho (K - 2) + 1]$.
> > > >
> > > > The third issue concerns $\boldsymbol \Lambda$ and the coefficients $\eta_x$. Indeed, $2 w_i \epsilon_0$ from the bound of $\Vert II_{1 1} \Vert_2$, $2(K - 1) \Delta (\epsilon_0 + \rho) w_\text{max}$ from the bounds of $\Vert II_{1 2} \Vert_2$ and $\Vert II_{1 3} \Vert_2$, and $2 (K - 1)^2 \Delta^2 (\rho + \xi) w_\text{max}$ from the (uncorrected) bound of $\Vert II_{1 4} \Vert_2$ are not present in $\boldsymbol \Lambda$. Moreover, in $\eta_2$, the term $2 \epsilon_0 \rho (K - 1) w_\text{max}$ from the bound of $\Vert II_{1 1} \Vert_2$ became $ \epsilon_0 \rho (K - 1) w_\text{max}$ and the term $\rho^2 (K - 1) w_\text{max}$ from the same bound became $\rho^2 (K - 2) w_\text{max}$ in $\eta_3$. This last mistake should not change anything.
> > > >
> > > > The fourth issue is just a question: how do you get the equation of line 91? I think that this is based on the fact that the numerator of $II_1 = \nu_0 w_i \mathbf C_{: i} + \boldsymbol \Lambda$ and that $\mathcal Y = \mathcal Y^* + \mathcal E$ but I did not manage to retrieve the bound, can you help me on that? I also think that there is a mistake, and in the following lines, $w_\text{max} \Vert \boldsymbol \Lambda \Vert_2$ should be replaced with $ \Vert \boldsymbol \Lambda \Vert_2$. I also was not able to retrieve the inequalities of line 94. Giving an upper bound of (K - 1) \Delta could help.
> > > >
> > > > The fifth issue is again a question: how do you handle the term $| \eta_0 - (1 - \epsilon_0^2) |$? Doing the computations, I find (13) from (12) by neglecting this term.
> > > >
> > > > The sixth issue: you conclude in line 100 with the desired upper bound + $\beta \epsilon_0$ ($\epsilon_0$ is missing, by the way) with $\beta \leq 1$. To get the final result, you have to iterate and need $\sum_n \beta^n < \infty$, but this is only true if $\beta < 1$. Can you prove that?
> > > >
> > > > I may have missed things or made mistakes, so do not hesitate if you disagree with the previous points.
> > > >
> > > >
> > > > Miscellaneous minor comments:
> > > > - line 71: there is still an extra equal sign in the definition of the spectral norm.
> > > > - line 121: $\mathbf u$ should be one of the arguments instead of $\mathbf u$ in the definition of the Fuse operator.
> > > > - line 147: to be consistent with line 96, $\mathbf i = [i, i + 1]$ should be replaced with $(i, i + 1)$.
> > > > - line 191: it is coherent now, but do you have an intuition why the lower bound is better if the weights are similar?
> > > > - line 209: just before Theorem 2, true cluster means are written $\boldsymbol \mu$, and in Theorem 2, they are written $\boldsymbol \mu^*$. Keeping the lighter version is probably better.
> > > > - in Table 1, the standard deviation is relatively high in the $\mu = 0.1$ column. Does this remain true with multiple random starts?
> > > >
> > > > - in $II_{14}$, $j_1$ should be used for $\mathbf A$ and $j_2$ should be used for $\mathbf B$.
> > > > - in the definition of $\Delta$, why is the second line true, and why do we need to replace $\mathbf A_{: i}$ with $\mathbf A_{: i} + \hat{\xi}_i$?
> > > > - Appendix line 72: forgetting that we are comparing vectors, a maximum over $j$ should appear.
> > > > - Appendix lines 77 and 78: sums should be over $l \neq i$.
> > > > - Appendix line 84: it corresponds to $ii_3$.

---

> > > > > ### Author Response · Authors · 2022-08-09
> > > > > **Thanks for the further comments!**
> > > > >
> > > > > We sincerely appreciate the constructive feedback from the reviewer which will lead to a much improved paper!
> > > > >
> > > > > **On the first issue** We apologize for the typos which are now corrected. As the reviewer noted, we plan to split $II_{12}$ into two parts, the first part is $\Lambda$ (in our original notation, $\Lambda$ is a scalar and we will modify it in the appendix, which should be $\Lambda = a\mathbf{1}$) which can be expressed as a bounded scalar $a$ times a vector of 1s with the conformable size of $\boldsymbol C_{:l}$; the second part is exactly what the reviewer describes $\nu_0w_l\boldsymbol C_{:l}$. Since, in the later analysis when $\Lambda$ appears, we only care about the upper bound on its $\ell_2$ norm (in equations 12, 13, 14), changing $\Lambda$ to bounded scalar $a$ times vector of 1 will not affect our final convergence results for factors $\boldsymbol C_{:l}, l>1$. We thank the reviewer for pointing out the unclear aspects in the proof of Theorem 1 and below are some updates we will revise in the appendix:
> > > > > 1. We revised the definition $II_{12} = (K-1)\Delta \Vert II_{12}^\prime\Vert_2\mathbf{1}+ (K-1)\Delta(\xi+\rho)w_i\boldsymbol C_{:i}$ with $\Vert\Delta II_{12}^\prime\Vert_2\leq \Vert i_1\Vert_2 +\Vert i_2\Vert_2 +\Vert i_3^\prime\Vert_2 + \Vert i_4^\prime\Vert_2$$\leq \xi\epsilon_0w_{\max}\alpha + (\epsilon_0+\xi)\rho(K-1)w_{\max} + \rho^2(K-2)w_{\max} + (\epsilon_0+\rho)w_\max$.
> > > > > 2. $\xi$ is upper bound for $\max_{j<i}\Vert \xi_j\Vert_2$ which is upper bounded by $\xi \leq 10\gamma\alpha K/\sqrt{d}$ (The result derived in Lemma 3).
> > > > >
> > > > > We hope the current version for the related parts in the proof is now clear.
> > > > >
> > > > > **On the second issue** In our bound, choices for $j_1$ and $j_2$ do not have a large effect on the final bound of $II_{14}$ since we take the uniform bound for $\sum_{j_1\leq i}\bar{\boldsymbol A}_{:j_1}^\top \boldsymbol A_{:i}$ and $\sum_{j_2\leq i}\bar{\boldsymbol B}_{:j_2}^\top \boldsymbol B_{:i}$, i.e. $(K-1)\Delta$.
> > > > >
> > > > > Thus, the relevant term in $ii_4$ that we care about is \max_{j_1<i, j_2<i}\mathcal{Y}^*(\boldsymbol A_{:j_1}, \boldsymbol B_{:j_2}, I). A detailed analysis could be: (1) if the maximum is attained when $j_1\neq j_2$, the upper bound for $\Vert ii_4\Vert_2$ is exactly what we derived in the appendix. (2) if the maximum is attained when $j_1 = j_2$, $\Vert ii_4\Vert_2\leq (K-1)\rho^2w_{\max} + w_{\max}\rho$. Since the difference between these two bounds is just the difference between $\rho^2 w_{\max}$ and $\rho w_{\max}$, it can be easily proved that $\rho^2 w_{\max} < \rho w_{\max}$ due to Assumption 1 where $\rho\leq \alpha/\sqrt{d}$ is an incoherence parameter which relaxes the orthogonalization condition on the latent factors. Thus, the upper bound on $\Vert ii_4\Vert_2$ under the scenario (1) $j_1\neq j_2$ is a loose bound compared to the scenario (2) $j_1 = j_2$. That's the reason we don't state the result of the second scenario.
> > > > >
> > > > > We will add a discussion on the bound of $\Vert ii_4\Vert_2$ in the final version of the appendix and thank the reviewer for pointing this out to allow us to make this important clarification!
> > > > >
> > > > > **On the third, fourth and fifth issues** We are grateful to the reviewer for raising these concerns and we apologize for not being clearer. Because of the close relationship of these three issues, we combine them and give some short answers here. We will add a more detailed derivation on upper bound of $\Vert \Lambda\Vert_2$ in terms of $\eta_x$, lower bound on the denominator of $II_{1}$ and final upper bound on $\Vert II_{1}\Vert_2$ in the revised appendix.
> > > > > 1. On the detailed definition of $\eta_x$, we have done some reorganization on $\eta_2, \eta_3$ using the relationship between $\epsilon, \rho, (K-1)\Delta, \xi$ which will make the derivations look clearer. Details will be added in the revised version.
> > > > > 2. In terms of Line 91, the short answer is that we follow the similar proof strategy as in step 1 (which is the convergence error bound for the first factor $\boldsymbol C_{:1}$, detailed procedure is in proof of Theorem 3 (step 2) in Sun & Li, 2019).
> > > > > 3. In terms of derivation for equation 13, we don’t neglect the term $\eta_0 - (1-\epsilon_0^2)$. The upper bound for $\ell_2$ norm of the whole numerator can be simplified as $2\Vert\Lambda\Vert_2 + (\eta_0+1+\epsilon_0^2)w_{\max} + \psi$. $2\Vert\Lambda\Vert_2\leq 2w_{\max}(\alpha\epsilon_0^2 + \epsilon_0\rho(K-1) + \rho^2(K-2))$ which is closely related to definition of $f(\epsilon_0, \rho, K)$ defined in appendix Line 34. Thus, the remaining work is to show $\eta_0 - (1-\epsilon_0^2)\leq 2(\alpha\epsilon_0^2 + \epsilon_0\rho(K-1) + \rho^2(K-2))$ which can be proved under the initialization condition on $\epsilon_0$.

---

> > > > > > ### Author Response · Authors · 2022-08-09
> > > > > > **Additional responses**
> > > > > >
> > > > > > **On the sixth issue** For the derivation of Line 100, the whole error bound can be split into 2 parts, the first part is in the order our final result of Theorem 1, which is $\gamma\rho^2(K-1) + \psi/w_{\min}$; the second part is $12\sqrt{2}w_{\max}/w_{\min}\tilde{q}$. Recall the defination of $\tilde{q}$ from Line 40, $\tilde{q} = \alpha\epsilon_0 + 2\rho(K-1)$. Thus, to show $12\sqrt{2}w_{\max}/w_{\min}\tilde{q} < 1$, it's equivalent to show $12\sqrt{2}w_{\max}/w_{\min}(\alpha\epsilon_0 + 2\rho(K-1))<1$. We transfer this as a condition on the initialization $\epsilon_0$, i.e., $\epsilon_0< 1/(12\sqrt{2}\gamma\alpha) - 2\rho(K-1)/\alpha$. We have included this initialization condition in Assumption 2 Line 168.
> > > > > >
> > > > > >
> > > > > > **On Line 191 intuition of $\gamma$**  Our intuition is as follows: as $\gamma$ increases, WLOG can assume $w_1\leq w_2 ... \leq w_K$, under the constraint that the spectral norm of $\mathcal{Y}^*$ is bounded, the minimum 'signal' $w_1$ is small which makes the latent factors difficult to recover (imagine the extreme case when $w_1\approx 0$, $w_1 \boldsymbol A_{:1}\circ \boldsymbol B_{:1}\circ \boldsymbol C_{:1}\approx 0$ and thus $\boldsymbol A_{:1}, \boldsymbol B_{:1}, \boldsymbol C_{:1}$ are difficult to detect). We plan to add more experiments in the revised manuscript to validate this with a goal of illustrating a different recovery error when $\gamma$ takes {large, medium, small} values.
> > > > > >
> > > > > > **On large standard deviation of $\mu=0.1$ in table 1** Our results stated in Table 1 are exactly the experimental results of 50 replications with multiple random starts. Our intuition of large standard deviation for $\mu = 0.1$ is that the distance between four clusters in $\boldsymbol C$ is smaller for $\mu=0.1$ compared to $\mu=0.7$, which increases the difficulty of clustering tasks.
> > > > > >
> > > > > > **On readability and typos**
> > > > > > The reviewer provided a lot of excellent suggestions for improving readability. We will use all of them in the revised version.

---

> ### Author Response · Authors · 2022-08-02
> **Additional references.**
>
> We sincerely appreciate the hard work of the reviewer who provided an extremely detailed and helpful report.
>
> *Additional references:*
>
> Anandkumar A, Ge R and Janzamin M (2014) Guaranteed Non-Orthogonal Tensor Decomposition via Alternating Rank-1 Updates. arXiv:1402.5180 .
>
> Sun WW, Lu J, Liu H, and Cheng G (2017) Provable sparse tensor decomposition. Journal of the Royal Statistical Society: Series B (Statistical Methodology), 79, 3, 899–916
>
> Sun WW and Li L (2019) Dynamic tensor clustering. Journal of the American Statistical Association, 1–28
>
> Tomioka R and Suzuki T (2014). “Spectral norm of random tensors.” arXiv:1407.1870
>
> Sharan V and Valiant G (2017). “Orthogonalized ALS: A theoretically principled tensor decom-
> position algorithm for practical use.” Proceedings of the 34th ICML, 3095–3104.

---

### Official Review · Reviewer_UQxe · 2022-07-12

**Rating:** 7
**Confidence:** 4
**Soundness:** 3 good
**Presentation:** 3 good
**Contribution:** 3 good

**Summary:**

This work proposes an algorithm for simultaneous tensor decomposition and clustering using the alternating least squares approach. The proposed method is accompanied by a complexity and convergence analysis. Experimental studies on both synthetic and real-world datasets demonstrate the effectiveness of the proposed method in clustering multiway array data.

**Questions:**

1. Why is a CP model chosen over other tensor decomposition models? Please elaborate.
2. Is the CP assumption relevant on real-world datasets?
3. The paper claims that employing orthogonalization in ALS avoids local minima and speeds up convergence. Although empirical evidence of this general statement it provided in previous works, does this hold in this setting where ALS is employed for clustering? Please provide intuition.



**Limitations:**

The paper discusses a couple of limitations of this work in section 6 and aims to address them in future work.

**Strengths And Weaknesses:**

Strengths:
1. Paper is well-written and easy to follow.
2. The proposed method is well motivated.
3. The proposed method is presented as a concise pseudocode and it is easy to follow.
4. Complexity analysis shows the proposed method's cost is similar to that of ordinary ALS, showing that it is practically applicable.
5. Section 3.2 on how to choose regularization weights shows that these vital hyperparameters are chosen systematically from the data and not heuristically.

Weakness:
1. Missing detailed intuition behind the reason for choosing a CP model as opposed to a Tucker model for example.

---

> ### Author Response · Authors · 2022-08-02
> **Thanks for the comments and hope our answers address your concerns**
>
> We thank the reviewer for the  constructive comments, positive feedback, and supportive rating.
>
> *On the intuition for choosing the CP model:*   We choose CP decomposition model based on the following considerations：
>
> 1. Compared to the Tucker decomposition, CP decomposition is unique under weaker conditions where by uniqueness, we mean for tensor $\mathcal{Y}\in\mathbb{R}^{n_1\times n_2\times n_3},$ there is only one possible combination of rank-one tensors $\sum_{r=1}^R a_r\circ b_r\circ c_r$ that sums to $\mathcal{Y}$, with the exception of elementary indeterminacies of scaling and permutation. Kruskal's result provided a sufficient condition for uniqueness and Ten Berge and Sidiropoulos (2002) showed the sufficient condition is also a necessary condition when $R=3$. Later papers considered more general necessary conditions that work for the case when $R>3$. On the contrary, Tucker decomposition is not unique. If we let $U,V,W$ be non-singular matrices, then $\mathcal{G}\times_1 A\times_2B\times_3 C = (\mathcal{G}\times_1 U \times_2 V \times_3 W)\times_1 AU^{-1}\times_2 BV^{-1}\times_3 CW^{-1}$. In other words, we can modify the core $\mathcal{G}$ without affecting the fit so long as we apply the inverse modification to the factor matrices.
> 2. Since the tensor Gaussian mixture model can be viewed as a special case of our proposed model (please refer to explanation on Line 91), we choose CP decomposition as backbone to uncover the clustering structure. Our method does not impose a distributional assumption on the error tensor $\mathcal{E}$. However, if assuming the error tensor follows a standard Gaussian distribution, then the proposed model reduces to a tensor version of the Gaussian mixture model, which enjoys the advantage that they do not require which subpopulation a data point belongs to and allows the model to learn the subpopulations automatically.
> 3. Lastly (but not least), we do not want to overemphasize the advantage of the CP decomposition method since there are lots of excellent multiway tensor clustering methods based on both decomposition models, CP and Tucker. In our opinion, it's not a yes or no question in terms of which decomposition method is implemented since they are just two different paths to explore clustering.
>
> We will add more discussion to this important point (choice of CP) in the final version of the manuscript.
>
> *On CP assumptions in real applications:*
>
> The assumptions we impose are common (and relevant) for the applications the methods are intended for. One example is the multi-tissue, multi-individual gene expression experiment where the data take the form of an order-3 tensor with three modes representing genes, individuals, and tissues. For simplicity, we denote the latent factors for genes, individuals, and tissues as $G_r, I_r, T_r$ respectively. The rank-1 component under CP assumption in this example $G_r\circ I_r\circ T_r$ can be interpreted as the basic unit of an expression pattern (called an expression module). Many papers on this genetics topic adopt this assumption and we list two of them: Hore et al. (2016), Wang et al. (2019). These additional references will be part of the final version of the manuscript.
>
> *On speed up convergence of orthogonalization:*
>
> ​​Intuitively, the periodic orthogonalization prevents multiple recovered factors from “chasing after” the same true factors, allowing for the avoidance of poor local optima and more rapid convergence to the true factors. Theoretically, we proved the upper bound for recovered factors, which enjoys consistency when some mild assumptions are imposed on error and $w_\min$ (Corollary 1). We also did several experiments to show the quicker convergence of adding orthogonalization. Results are provided in Figure 2(d) where the experiments are based on the comparison of dynamic tensor clustering algorithm (which adopts the similar CP decomposition assumption, but the algorithm does not contain orthogonalization step) and our proposed method. That brings clear evidence that adding orthogonalization speeds up the convergence. We thank the reviewer for raising this issue.
>
> *Additional references:*
>
> Kruskal JB (1977) Three-way arrays: rank and uniqueness of trilinear decompositions, with application to arithmetic complexity and statistics. Linear algebra and its applications, 18(2), 95-138.
>
> Kruskal JB (1989). Rank, decomposition, and uniqueness for 3-way and N-way arrays. In Multiway data analysis (pp. 7-18).
>
> Ten Berge JM & Sidiropoulos ND (2002). On uniqueness in CANDECOMP/PARAFAC. Psychometrika, 67(3), 399-409.
>
> Hore V, Vinuela A, Buil A, Knight J, McCarthy MI, Small K, & Marchini J (2016). Tensor decomposition for multiple-tissue gene expression experiments. Nature Genetics, 48(9), 1094-1100
>
> Wang M, Fischer J & Song YS (2019). Three-way clustering of multi-tissue multi-individual gene expression data using semi-nonnegative tensor decomposition. The Annals of Applied Statistics, 13(2), 1103.

---

> > ### Comment · Reviewer_UQxe · 2022-08-10
> > **Response to author reply**
> >
> > Authors, thank you for the detailed response.
> > I am satisfied with the explanation provided for my questions. I stick to my original rating.

---

### Author Response · Authors · 2022-08-02
**Thank you!**

We thank all reviewers for their detailed feedback and constructive suggestions! They will contribute to a much improved final version.

We appreciate all comments pointing out insufficiencies and providing ideas for future improvements. Below we answer each reviewer’s questions and issues raised as separate comments.

---

### Meta-Review · Area_Chair_jYza · 2022-08-24

**Recommendation:** Accept
**Confidence:** Certain

**Metareview:**

The paper proposes a multi-mode tensor clustering method using an alternating least square algorithm. The reviewers unanimously like the paper and the content. Some reviewers have sought a few clarifications including on the proofs. Hope the authors would address them in the revised version.

**Award:**

No

---

### Decision · Program_Chairs · 2022-09-14

Accept